# Sensory Modulation in Children with Developmental Coordination Disorder Compared to Autism Spectrum Disorder and Typically Developing Children

**DOI:** 10.3390/brainsci12091171

**Published:** 2022-08-31

**Authors:** Sofronia M Ringold, Riley W McGuire, Aditya Jayashankar, Emily Kilroy, Christiana D Butera, Laura Harrison, Sharon A Cermak, Lisa Aziz-Zadeh

**Affiliations:** 1Mrs. T.H. Chan Division of Occupational Science and Occupational Therapy, University of Southern California, Los Angeles, CA 90089, USA; 2Brain and Creativity Institute, Dornsife College of Letters, Arts, and Sciences, University of Southern California, Los Angeles, CA 90089, USA; 3Department of Pediatrics, Keck School of Medicine, University of Southern California, Los Angeles, CA 90033, USA

**Keywords:** developmental coordination disorder, sensory processing, autism spectrum disorder, behavior, social emotional, motor skills, dyspraxia, empathy, sensory modulation, anxiety

## Abstract

Developmental Coordination Disorder (DCD) is one of the least studied and understood developmental disorders. One area that has been minimally investigated in DCD is potential issues with sensory modulation. Further, in other neurodevelopmental disorders (e.g., autism spectrum disorder (ASD)) sensory modulation is related to many other challenges (e.g., social issues, repetitive behaviors, anxiety); however, such potential relationships in children with DCD have been largely unexplored. The purpose of this study is to explore sensory modulation differences in DCD and to understand the relationships between sensory modulation and social emotional measures, behavior, and motor skills in DCD in comparison to ASD and typically developing (TD) peers. Participants (aged 8–17) and their caregivers (DCD, *N* = 26; ASD, *N* = 57; and TD, *N* = 53) completed behavioral and clinical measures. The results indicated that 31% of the DCD group showed sensory modulation difficulties, with the DCD group falling between the ASD and TD groups. In the DCD group, sensory modulation was significantly associated with anxiety, empathic concern, repetitive behaviors, and motor skills. Data are compared to patterns seen in ASD and TD groups and implications for interventions are discussed.

## 1. Introduction

Sensory processing is defined as “the ability to manage detection, modulation, interpretation, and organization of incoming sensory information” [1,2]. Recently, there has been an explosion of interest in sensory processing, in part related to the addition of sensory responsivity to the diagnostic criteria of autism spectrum disorder (ASD) [3] and the increasing recognition that impairments in sensory processing, sometimes referred to as sensory processing disorder (SPD), affect everyday performance and participation [1,4,5,6,7,8]. Sensory processing includes both the ability to discriminate sensory information as well as the ability to modulate incoming sensory information. Individuals with problems in sensory discrimination have difficulty identifying the temporal and spatial features of sensory input, which may impact body awareness, motor planning, and motor skills [9,10,11]. In comparison, challenges with sensory modulation may impact the ability to perceive and respond to certain stimuli in a regulated manner appropriate for the context [2,8]. In addition, sensory processing patterns have been established which categorize behavioral responses to sensory stimuli [12]. 

Impairments in sensory processing have been identified in children with various neurodevelopmental disorders including attention hyperactivity deficit disorder (ADHD) [2,13,14], ASD [9], Williams Syndrome [15], and preterm infants [16], with several investigators focusing on the identification of the sensory processing differences between clinical populations [17,18,19,20]. 

### 1.1. Developmental Coordination Disorder (DCD)

Developmental Coordination Disorder (DCD), also known as dyspraxia, is a neuromotor disorder characterized by impairments in the development of motor coordination including dexterity, limb speed, and gross and fine motor skills that are not due to an intellectual disability or other neurological disorder affecting movement [3,21,22]. It is estimated that DCD affects 5–6% of children and may significantly interfere with activities of daily living and school performance [21,23]. DCD is also a common comorbidity in children with ASD, with approximately 60–90% meeting the criteria for DCD [24,25,26]. However, 90% of children identified with DCD do not have a co-occurring diagnosis of ASD [27].

### 1.2. Sensory Processing in Developmental Coordination Disorder

#### 1.2.1. Sensory Discrimination

Many studies have reported differences in sensory processing, specifically somatosensory (tactile and kinesthetic) discrimination, in children with DCD or probable DCD (those without a formal DCD diagnosis) and have identified a positive relationship between somatosensory discrimination and motor planning, motor learning, and motor skills [9,28,29,30,31,32,33,34,35]. Ayres [9,28,36] theorized that somatosensory discrimination and its integration with visual and vestibular processing is important to the development of an adequate body schema, which is critical for motor planning. These proposed relationships were supported in more recent studies by Mailloux et al. [37], Mulligan [38,39], and Roley et al. [40].

#### 1.2.2. Sensory Modulation

In comparison to the number of studies examining tactile–kinesthetic discrimination in children with DCD, fewer studies have examined sensory modulation. A review of these studies indicates that many children with DCD also show differences in sensory modulation, including hypersensitivity to tactile, auditory, visual, and olfactory stimuli, though not to the same extent as children with ASD [18,41]. Using the Sensory Processing Measure (SPM) parent questionnaire, Allen and Casey [41] reported that most children with DCD (88%) in their sample had scores in the definite difference range indicating difficulties in sensory processing across domains of the SPM (social participation, vision, hearing, touch, body awareness, balance and motion, planning and ideation) that may have a noticeable effect on their child’s daily activities. However, 46% of their DCD sample had a codiagnosis of ASD, a population known to have a very high prevalence of sensory modulation differences, as mentioned earlier. These findings indicate that, whereas sensory processing differences are much more common in ASD, children with DCD may also have a high prevalence of sensory processing impairments that need to be further explored.

#### 1.2.3. Sensory Processing Patterns

There is evidence that children with DCD display different sensory processing patterns when compared with other clinical populations or TD children [42,43]. For example, Delgado-Lobete et al. [42] used the Short Sensory Profile-2 (SSP-2) [12] and reported that at least one atypical sensory processing pattern was present in 45% of children with DCD, significantly higher than in the TD group (25.7%) but lower than the ADHD group (85.2%). For children with DCD, the most common pattern was avoider (28.3%), which is described as individuals who have a low sensory threshold and engage in behaviors to counteract sensory stimulation. The second most common pattern was bystander (26.1%), defined as individuals with a high sensory threshold, who miss more sensory cues than usual. The least common pattern in the DCD group was seeker (19.6%), characterized as individuals who have a high sensory threshold and look for environments rich in sensory stimuli. The sensor pattern, described as those with a low sensory threshold, who display behaviors to avoid discomfort with sensation, was the most common in the ADHD group (77.8%) followed by the DCD group (21.7%) and was least common in the TD children (9.5%). In a study on children with ASD, Simpson and colleagues [44] found elevated scores in all four sensory processing patterns, with avoider and sensor (hypersensitivity) the most common. Overall, these studies indicate that children with DCD have greater atypical sensory modulation compared to TD children but less than children with ASD.

### 1.3. The Relationship between Sensory Processing and Social Emotional Measures

#### 1.3.1. Social Skills

Numerous studies have reported an association between sensory processing and social participation in different clinical populations of children including SPD [45], ASD [46,47,48,49], and ADHD [14,50]. In children with ASD, sensory over-responsivity (SOR) has been correlated with difficulties with social engagement, communication, and interaction [51]. In a systematic review, Koenig and Rudney [52] reported that children and adolescents with DCD that exhibit sensory over-responsive patterns and sensory–motor dysfunction have more difficulty with social participation than children with sensory under-responsivity. The relationship between sensory processing and social abilities in children with DCD/dyspraxia has been less explored than in those with ASD. While in ASD, social impairment is a key diagnostic criterion, recent studies indicate that children with DCD also commonly have social deficits including social anxiety, less peer support, and poor social communication [21,53]. In fact, our prior work showed that about 36% of children with DCD fell into the clinical range of social deficits on the SRS-2 [25]. Although these social issues are often described as secondary symptomologies, derived from having less social peer interactions by not participating in social activities that require motor skills such as sports teams, instrumental groups, or art classes [54,55], an alternative interpretation is that some social symptoms may be primary in DCD.

#### 1.3.2. Anxiety and Depression

Extensive research has shown that anxiety and depression are commonly reported in children with ASD [56,57,58] and that anxiety is associated with sensory over-responsivity [57,59]. Although less reported, an increasing number of articles have begun to examine anxiety and related mental health issues in individuals with DCD [60]. For example, parents of children with DCD reported significantly greater anxiety and more depression in their children than did parents of TD children [61,62,63,64]. Previous studies on sensory processing and depression/anxiety have mainly focused on adults or children with ASD and have mixed results. Liss et al. [65] found a positive correlation between hypersensitivity and depression and hypersensitivity and anxiety in young adults. Neal et al. [66] found that hypersensitivity was a predictor of anxiety but not depression in adults with anxiety or depression. Thus, further research needs to be done to understand the association between sensory processing and anxiety/depression in clinical populations to help inform interventions [67].

#### 1.3.3. Alexithymia

Alexithymia is a subclinical feature characterized by difficulty recognizing, describing, and distinguishing between one’s own emotions. It is very common in ASD (~50%) [68,69] and has been studied in other clinical groups including mood disorders and anxiety [65,70]. Prior studies show that atypical sensory profiles are related to increased alexithymia in ASD and TD samples [65,69,71,72]; however, to our knowledge, the prevalence of alexithymia and its relationship to sensory processing in DCD has yet to be established.

#### 1.3.4. Empathy

Recent studies have explored a connection between sensory processing and empathy [23,73]. Researchers have also identified links between interoceptive abilities and empathy in typically developing [74,75] and autistic individuals [73,76,77], though few have examined empathy and sensory processing (exteroception). It has been suggested that individuals who have high sensory responsivity experience higher emotional reactivity, display more empathy, and more activation in brain regions associated with empathy (i.e., cingulate and insula) than those with lower sensory sensitivities [78,79]; however, Schaefer et al. [80] did not find correlations with high sensory sensitivity and empathy in TD adults. Although, most of the research supporting this claim has been performed in typically developing adults. Tavassoli et al. [81] found that parents rated their children with ASD as having less empathy than did parents of children with SPD and TD children. In addition, empathy was negatively correlated with sensory over-responsivity symptoms. To our knowledge, such comparisons in DCD have not been performed.

### 1.4. The Relationship between Sensory Processing and Behavior

Miller et al. [7] examined adaptive behaviors associated with social emotional functioning, attention, and dyspraxia among children with different patterns of sensory processing. They found that children with high sensory responsivity (over or under) displayed more problem behaviors, such as more aggressive behaviors and hyperactivity, than those with typical sensory processing patterns. Their results also showed the coexistence of sensory modulation symptoms (over or under responsivity, sensory craving), social emotional issues (depression, anxiety), and praxis problems. Cosbey et al. [45] reported that children with sensory processing disorder displayed more frequent peer conflict, less responses to social cues, and were sought out less than typically developing peers on the playground. Additionally, in children with ASD, sensory over-responsivity has been shown to be correlated with repetitive behaviors [82,83], but this relationship has not been examined in children with DCD. In summary, there is evidence that, similar to children with ASD, associations between sensory processing and behavior may be present in children with DCD, especially those with sensory processing differences. 

### 1.5. The Relationship between Sensory Processing and Motor Skills

Previous research has shown that differences in sensory processing (as measured by the SSP) is associated with impaired motor skills in children with ASD [84,85] and that sensory processing interventions, such as sensory integration, may improve motor performance [14,86,87]. Although this relationship has not been established in DCD, we expect a similar association due to similar motor deficits in children with ASD and DCD.

### 1.6. Current Study 

In prior studies on sensory processing in DCD, either: (a) DCD was compared to neurotypical children [42,43], (b) a retrospective chart review was performed [41], or (c) DCD was compared to ASD only [18]. To our knowledge, only one previous study from our group examined relationships between motor and sensory processing in DCD, ASD, and TD [23]. Here we build on our prior data by comparing the relationship between sensory processing and social emotional behavior in all three groups (TD, ASD, DCD). 

The purpose of the current study was to explore the contributions of sensory modulation to the DCD phenotype and to understand the relationship between sensory, social emotional, behavior, and motor features in this population in comparison to children with ASD and TD children. We expand upon our previous findings [23] by including larger sample sizes, a different socio-emotional measure (e.g., SCARED-P), and an assessment of repetitive behavior (RBS-R). The following questions are addressed:Do children with DCD experience differences in sensory processing? How does this compare to ASD and TD peers? Based on our previous work [23], we predict children with DCD to fall between ASD and TD groups in sensory over-responsivity. We further expand our prior studies by investigating specific aspects of sensory modulation and determine how many children in each group meet clinical cutoffs.Do children with DCD differ from children with ASD and TD children on social emotional and behavioral measures? Based on our prior data [23,25], we expect children with DCD to fall between ASD and TD groups on social emotional measures, competencies, and empathic skills. We also explore differences in DCD that have previously not been explored in our prior data: anxiety as measured by the SCARED-P and repetitive behaviors.How does sensory over-responsivity in children with DCD correlate with social emotional, motor, and behavioral measures? Do these relationships differ in DCD as compared to ASD and TD children? We expect significant correlations between SOR and anxiety and the CBCL in all groups, significant correlations between SOR and motor difficulties in the ASD and DCD group, and significant correlations between SOR and social difficulties in the ASD group.

## 2. Materials and Methods

### 2.1. Participants

The Institutional Review Board approved this study at the University of Southern California (USC). A total of 136 right-handed individuals aged 8–17 in either DCD (*N* = 27; mean age = 11.75, SD = 2.31), ASD (*N* = 57; mean age = 11.89, SD = 2.29), or TD (*N* = 53; mean age = 11.75, SD = 2.13) groups completed the study. This study was part of a larger study that included brain imaging components, thus some inclusion criteria were based on those requirements [88]. Using the effect sizes from a meta-analysis on TD and ASD by Ben Sasson et al. [89], we conducted a power analysis and found that the sample sizes we had were sufficient to achieve 97.5% power. Additional power analysis based on previous work in our lab on pairwise comparisons for sensory modulation in ASD, TD, and DCD groups [23], showed that sample sizes of at least 25 per group would be suitably powered based on the effect sizes for the strongest predictors for group membership.

Participants were recruited from clinics in the greater Los Angeles healthcare system, through local schools, word-of-mouth, and social media advertising. Inclusion criteria for all participants included: (a) IQ of at least 75 on either the Full-Scale Intelligence Quotient (FSIQ) or Verbal Comprehension Index (VCI) of the Wechsler Abbreviated Scale of Intelligence, 2nd Edition (WASI-II) [90]; (b) right-handed, as assessed by a questionnaire adapted from Crovitz and Zener [91]; (c) fluent in English, with at least one parent fluent in English. Exclusion criteria for all participants included: (a) history of head injury with loss of consciousness greater than 5 min; (b) born before 36 weeks of gestation; (c) contraindications to participating in MRI. All participants and parents were evaluated for their capacity to give informed consent and then provided their written child assent and parental consent in accordance with the study protocols approved by the university’s Institutional Review Board.

The DCD group consisted of 26 participants (11 female, 15 male). Eligibility criteria for the DCD group included: (a) performance at or below the 16th percentile on the MABC-2, (b) no first-degree relatives with ASD, and (c) no current or previous concerns about an ASD diagnosis. The Conners 3AI-Parent report was used to identify ADHD symptoms but was not used as an exclusion criterion since ADHD is highly comorbid with DCD [92]. Seven children in the DCD group who had a T-score range of 65–74 on the SRS-2 were administered the ADOS-2 but did not meet the criteria for ASD and were thus included in the study. Five DCD participants were taking prescribed psychotropic medication at the time of data collection.

The ASD group consisted of 57 participants (13 female, 44 male). Additional inclusion criteria for the ASD group included a previous ASD diagnosis either through a clinical ASD diagnostic interview, an ASD diagnostic assessment, or both. Diagnosis was reassessed by research-reliable staff using the Autism Diagnostic Observation Schedule, Second Edition (ADOS-2) [93] and the Autism Diagnostic Interview-Revised (ADI-R) [94]. Two female participants had subthreshold ADOS-2 scores but qualified based on the ADI-R and by clinician review. All other participants met criteria on both the ADOS-2 and the ADI-R. Additional exclusion criteria included a diagnosis of other neurological or psychological disorders except for attention deficit disorders or generalized anxiety disorder, because those are highly comorbid with ASD [95]. Twenty ASD participants were taking prescribed psychotropic medication at the time of data collection.

The TD group consisted of 53 participants (22 female, 31 male). TD controls were excluded if they had: (a) any psychological diagnosis or neurological disorder, including attention deficit disorders or generalized anxiety disorder; (b) a first-degree relative with ASD; (c) a T-score above 65 on the Conners 3AI-Parent report [96], indicating a risk for attention deficit and hyperactivity disorder (ADHD); (d) a score below the 25th percentile on the Movement Assessment Battery for Children (MABC-2) [97] or probable DCD based on the Developmental Coordination Disorder Questionnaire (DCDQ) [98,99]; and (e) a T-score above 60 on the Social Responsiveness Scale, Second Edition (SRS-2) [100], indicating a risk for ASD. No TD participants were taking prescribed psychotropic medication at the time of data collection.

For all groups, sensory processing was measured using the SSP-2 [12] and the SensOR Inventory [101], and motor skills were measured using the DCDQ [98] and MABC-2 [97]. Social emotional skills were assessed using the SRS-2 [100], NEPSY-II [102], IRI [103], SCARED-P [104], and the AQC [105]. Behavior was assessed using the RBS-R [106] and CBCL [107]. 

### 2.2. Assessments

All assessments are described in detail below. Study data were collected and managed using Research Electronic Data Capture (REDCap) tools hosted at USC [108,109].

#### 2.2.1. Screening Measures

##### Wechsler Abbreviated Scale of Intelligence (WASI-II)

The WASI-II [90] is a measure of intelligence normed for ages 6–90 and contains four subtests including Block Design, Vocabulary, Matrix Reasoning, and Similarities. The Verbal Comprehension Index (VCI) consists of Vocabulary and Similarities subtests, and the Perceptual Reasoning Index (PRI) consists of Block Design and Matrix reasoning subtests.

##### Autism Diagnostic Observation Schedule (ADOS)

The ADOS-2 [93] is a semi-structured observational assessment that provides a standardized measure of social affect, ability to communicate, and restricted and repetitive behaviors. Module 3 (verbally fluent children and young adolescents) or Module 4 (verbally fluent older adolescents and adults) were administered by a trained researcher. The diagnostic exam provides a comparison severity score ranging from 1 to 10 (10 being the most severe) [93].

##### Autism Diagnostic Interview-Revised (ADI-R)

The ADI-R [94] is a clinical diagnostic instrument for assessing autism for ages 2 and up. It is a structured interview with the parent with open-ended questions and scored across three domains: Language/Communication (LC), Reciprocal Social Interactions (RSI), and Restricted, Repetitive, and Stereotyped Behaviors (RRB). 

##### Conners 3rd Edition ADHD Index—Parent (Conners 3AI)

The Conners 3AI [96], is a parent measure for children aged 6–18 years and consists of 10 items from the larger Conners scale that best differentiate children with and without ADHD. Items are rated on a 4-point Likert scale ranging from (0) “not true at all” to (4) “very much true”. It is age-normed and used to identify if an evaluation of ADHD might be necessary. A T-score below 60 is considered an average and scores of 65–90 are considered clinically significant.

#### 2.2.2. Sensory Measures

##### Short Sensory Profile-2 (SSP-2)

The SSP-2 [12] is a 34-item questionnaire designed to assess a child’s sensory processing patterns, including seeking/seeker (7 items), sensitivity/sensor (10 items), registering/bystander (8 items), and avoiding/avoider (9 items). The measure is for caregivers of children aged 3–14 years and includes items demonstrating the highest discriminative power of abnormal sensory processing from the items on the Sensory Profile-2 [12]. The items are rated on a 5-point Likert scale ranging from (5) “almost always” to (1) “almost never” and includes an option to select (0) “does not apply.” Raw scores from each category are totaled to determine where the child falls within each quadrant (registration, seeking, sensitivity, avoiding) ranging from “much less than others” to “much more than others.” A raw score falling one standard deviation outside of the mean typical range indicates potential clinical relevance.

##### Sensory Over-Responsivity Scale (SensOR)

The Sensory Over-Responsivity (SensOR) Inventory [101] is a valid and reliable 76-item questionnaire assessing the presence and severity of sensory sensitivity in individuals aged 3 and older. Parents respond to statements about their child’s preferences with either “Yes” or “No.” Subdomains include visual (5 items), olfactory (5 items), auditory specific (12 items), auditory settings (8 items), tactile daily living (19 items), tactile texture (9 items), food (9 items), and movement proprioception (9 items). Raw scores from all subdomains are added to get a total score. Higher scores indicate greater sensory sensitivity [101]. In addition to the total score, several researchers have calculated a tactile and an auditory score [110]. The sum of raw scores from the tactile domains (garments, self-care, and tactile sensations, 28 items) results in a tactile subscale score, and the sum of auditory domains (auditory specific and auditory settings, 20 items) are used for an auditory subscale score. 

#### 2.2.3. Motor Measures

##### Movement Assessment Battery for Children Second Edition (MABC-2)

The MABC-2 [97] assesses fine and gross motor performance skills and consists of three subsections: manual dexterity, aiming and catching (ball skills), and balance (static and dynamic). The Age Band 2 form was used for ages 7–10 and Age Band 3 was used for ages 11–16. The total standard score was used to determine percentile range. Percentile ranges < 5th indicate significant motor difficulty; a percentile range of 5–15th indicates the child may be at risk of motor difficulties; and a percentile range > 15th indicates no movement difficulty [97]. 

##### Developmental Coordination Disorder Questionnaire (DCDQ)

The DCDQ [98] is a tool used to screen for coordination disorders in children aged between 5 and 15. Parents rate 17 positively stated items (e.g., “My child can…”) on a 5-point Likert scale ranging from (1) “Not at all like your child” to (5) “Extremely like your child”, comparing their child’s coordination to same-aged peers. Scores fall between 17 and 85, with total raw scores of 0–48, 49–57, and 58–85 resulting in “an indication of DCD”, “suspect DCD”, and “probably not DCD”, respectively [98]. Subscales include “control during movement”, “fine motor/handwriting”, “gross motor/planning”, and “general coordination”. Higher scores indicate better motor coordination.

#### 2.2.4. Social Emotional Measures

##### Social Responsiveness Scale, 2nd Edition (SRS-2)

The SRS-2 [100] is a 65-item rating scale that has been designed to be used both as a screener and as an aid to clinical diagnosis for ASD. The school-aged form for ages 4–18 was completed by the parent. Items are rated on a 4-point Likert scale ranging from (1) “not true” to (4) “almost always true.” The questionnaire probes core diagnostic symptoms of ASD including social communication, restrictive interests, and repetitive behaviors and deficits in reciprocal social interactions. SRS-2 total T-scores between 60 and 65 are considered mild, scores between 66 and 75 are considered moderate, and scores over 75 are considered severe. 

##### A Developmental NEuroPSYchological Assessment—Second Edition (NEPSY-II)

The NEPSY-II [102] is a comprehensive neuropsychological measure for children and adolescents with 32 subtests across 6 domains (attention and executive functioning, language, memory and learning, sensorimotor, social perception, and visuospatial processing). The Affect Recognition (AR; 35 items) and Theory of Mind (ToM; 21 items) subtests within the social perception domain were utilized. The ToM subtest contains a verbal and contextual score. The sum of raw scores from each subtest are used to inform a percentile rank, scaled scores, and a social perception total score. Higher scores indicate greater ability. 

##### Interpersonal Reactivity Index (IRI)

Empathy skills were assessed using the IRI [103], a 28-item self-report measure consisting of four 7-item subscales: two cognitive empathy scales (perspective taking and fantasy) and two emotional empathy scales (empathetic concern and personal distress). The IRI has been validated for children as young as second grade [111]. Children rate statements about their thoughts and feelings on a 5-point Likert scale ranging from (1) “does not describe me well” to (5) “describes me very well.” A modified version with child-appropriate language was used [112]. Raw scores are added for each subscale score and higher scores indicate greater empathy.

##### The Screen for Child Anxiety Related Emotional Disorders-Parent (SCARED-P)

The SCARED-P [104] is a 41-item questionnaire administered as a parent-reported measure of anxiety symptoms in children aged 8–18. Items are presented on a 3-point Likert scale including (0) “not true or hardly ever true”, (1) “somewhat or sometimes true”, and (2) “very true or often true” [113]. The five subscales (generalized anxiety symptoms, separation anxiety symptoms, social anxiety symptoms, panic or somatic symptoms, school avoidance) are scored by adding the sum of each answer, which is combined to get a total score. A total score of ≥25 may indicate the presence of an anxiety disorder while there are individual cutoff scores for each subscale [114,115]. 

##### Alexithymia Questionnaire for Children (AQC)

The AQC [105] is a 20-item self-administered questionnaire representing 3 factors of alexithymia including difficulty identifying feelings, difficulty describing feelings, and externally oriented thinking. The AQC was standardized for children aged 9–15 [105]. Items are measured using a 3-point Likert scale ranging from 0 to 2: “not true”, “sometimes true”, and “often true”. The measure was partially based on the Toronto Alexithymia Questionnaire (TAS-20) for adults [105]. The third subscale, externally oriented thinking, was not used in this study due to an unacceptably low Cronbach’s alpha in the original psychometric paper and because alexithymia can be reliably assessed in youth without the eight items rating the externally oriented thinking (EOT) dimension [116]. Raw scores of each subscale and a 2-factor total were added to get factor totals. 

#### 2.2.5. Behavioral Measures

##### Repetitive Behaviors Scale-Revised (RBS-R)

The RBS-R [106] is a validated 43-item questionnaire that measures repetitive behaviors in children and adolescents aged 6–17 [117,118]. Parents rate items on a 4-point Likert scale ranging from (0) “behavior does not occur” to (3) “behavior occurs and is a real problem” within the past month [119]. The RBS-R assesses behaviors across 6 subscales (stereotyped behavior, self-injurious behavior, compulsive behavior, ritualistic behavior, sameness behavior, and restricted interests). The sum of the raw scores is used to calculate subscores and the total score. Previous research recommends focusing on the subscores, as opposed to the total score, to capture the nuances of such behaviors [120,121]. Higher scores indicate higher frequency of repetitive behaviors [106]. 

##### Child Behavior Checklist/6-18 (CBCL)

The CBCL/6-18 [107] is a norm-referenced, standardized assessment for children aged 6-18, assessing the likelihood of emotional, behavioral, and social problems related to a variety of psychological diagnoses [122]. Parents report on 20 items related to their child’s deficits and strengths in four competency areas (activities, social relations, school performance, and total competence) and 120 behavioral and emotional statements rated on a 3-point Likert scale ranging from (0) “not true” to (2) “very true or often true” [123]. The CBCL also includes 8 syndrome scales (thought problems, attention problems, social problems, anxious/depressed, withdrawn/depressed, somatic complaints, rule-breaking behavior, and aggressive behavior) [124]. T-scores of <65 are considered “normal”, scores of 65–69 are considered “borderline”, and scores ≥ 70 are considered “clinical”. 

### 2.3. Analysis

The data were analyzed using IBM SPSS Statistics (28.0, Armonk, NY, USA) for Macintosh (Cupertino, CA, USA). Descriptive statistics were calculated for the data set and are included in Table 1. One-way analysis of variance (ANOVA) was used to evaluate the differences between groups (TD, ASD, DCD) for age and IQ. Multiple comparisons correction was performed using the Bonferroni method when comparing the estimated marginal means for each group (Bonferroni, 1936*). Pearson’s Chi-squared test was run to evaluate group differences on gender. Univariate analysis of covariance (ANCOVA) was used to evaluate differences between groups in relation to covariates (age, sex, FSIQ) and measures of sensory processing (SSP-2 and SensOR Inventory), social emotional measures (SRS-2, NEPSY-II, IRI, SCARED-P, AQC, CBCL Syndromes), behavioral measures (CBCL competencies, RBS-R), and motor skills (DCDQ). Group comparisons were not performed for the SRS-2 as it was part of the inclusion criteria for the TD group. Scatterplots, boxplots, and histograms were generated for all variables and visually inspected to ensure normality and linearity.

#### 2.3.1. Determining Sensory Sensitivities

To determine sensory sensitivities in each group, we used a total score of two standard deviations above the mean of the TD group from Schoen and colleagues (2008 *) (≥14) on the SensOR Inventory. On the SSP-2, unusual sensory response patterns were defined as scores one or more normalized standard deviations above the mean (“more than others” and “much more than others”) for each pattern [12].

#### 2.3.2. Correlation Analyses

Partial correlations were run controlling for age, FSIQ, and gender. For correlational analyses, we focused on the SensOR Inventory total score rather than the SSP due to its validity and reliability to detect sensory over-responsivity [101]. To restrict the number of comparisons, we focused on the total scores for each measure except for the repetitive behavior scale due to studies suggesting greater validity when analyzing the subscores independently [120,121].

## 3. Results

### 3.1. Group Differences 

Between-group differences, including mean, standard deviations, effect sizes, and p-values for each group on each measure can be found in Table 1. Cohen’s d effect size from paired comparisons between subject groups are provided in the Appendix A. We report below the prominent findings as well as percentages by group that met clinical cutoffs. 

#### 3.1.1. Sensory Over-Responsivity (SensOR Inventory)

DCD and TD children did not display significant differences in sensory over-responsivity and scored significantly lower than children with ASD. Following the cutoff for sensory over-responsivity suggested by Schoen et al. [101], 31% of the DCD group, 74% of the ASD group, and 4% of the TD group qualified for SOR. Refer to Figure 1 for histograms of SensOR Inventory auditory, tactile, and total scores.

#### 3.1.2. Sensory Response Patterns (SSP-2)

On three of the four quadrants (seeker, avoider, sensor), the DCD group had significantly different scores from both ASD and TD groups, again falling between these groups. For the bystander pattern, the DCD and ASD groups both significantly differed from the TD group but did not significantly differ from each other. Of children with DCD in our sample, 37% demonstrated at least one unusual sensory response pattern with 46% bystanders, 31% sensors, 23% avoiders, and 19% seekers, compared to the TD group in which 11% demonstrated an atypical sensory pattern, with 11% bystanders, 6% sensors, 8% avoiders, and 6% seekers. Overall, 91% of the ASD group displayed at least one sensory response pattern “more than others”, with sensor the most common (74%), followed by avoider (67%), bystander (65%), and seeker (35%). The frequencies of sensory response patterns by group can be found in Figure 2.

#### 3.1.3. Anxiety

Parents of children with ASD reported significantly higher anxiety in their children than did parents of DCD and TD children, who were not significantly different from each other. Scores in the clinical range for anxiety disorder were found in 32% of ASD, 8% of DCD, and 2% of TD children. 

### 3.2. Correlation Analysis

Within-group partial correlation (r) coefficients can be found in Table 2 and Table 3. Refer to Figure 3 for scatterplots of significant correlations. Consistent with prior reports, the ASD group showed significant correlations between sensory over-responsivity and all our main social emotional measures: anxiety, social skills, empathy (perspective taking), alexithymia (describing feelings), repetitive behaviors, and CBCL syndromes (all *p*s < 0.05). The DCD group showed similar significant patterns with anxiety, domains of repetitive behaviors (self-injurious, sameness, total), empathy (empathetic concern), depression/emotional withdrawal (all *p*s < 0.05), and subthreshold relationships with social skills (*p* < 0.065). The DCD group additionally showed a significant relationship between SOR and motor impairment (*p* < 0.05). The TD group showed a significant relationship between SOR and anxiety and depression/emotional withdrawal.

## 4. Discussion

Developmental Coordination Disorder is one of the least studied and understood developmental disorders [125]. While there is evidence identifying impairments in somatosensory discrimination in children with DCD, there is limited investigation examining sensory modulation in this population. We discuss: (1) sensory processing/modulation differences in our DCD, ASD, and TD groups; (2) prominent differences in social emotional measures and behavior across groups; and (3) correlations between sensory processing and social emotional measures, behavior, and motor skills.

### 4.1. Group Differences

#### 4.1.1. Sensory Modulation in DCD

Our study found that children with DCD had significant sensory modulation differences in many, but not all, measures of sensory processing when compared to TD children and children with ASD. Consistent with previous studies [41], children with ASD had significantly more sensory over-responsivity than TD and DCD groups. Specifically, using the SensOR Inventory, we found that 31% of the DCD group, 74% of the ASD group, and 4% of the TD group qualified for SOR. Our results are consistent with the findings from Mikami et al. [43] and Delgado-Lobete et al. [42] which showed atypical sensory processing patterns in children with DCD. Here we expand upon Mikami et al.’s [43] findings and show similar patterns for children aged 8–18 and additionally find a significant difference between DCD and TD groups for the sensory seeking pattern. Furthermore, we found significant group differences between DCD and ASD on seeking, avoiding, and sensor, which have not been previously reported in the literature.

Interestingly, the DCD group did not differ from the ASD group on the bystander pattern (sometimes called hyporesponsivity or low registration), but both groups differed significantly from the TD group. Overall, these data indicate that there is at least a subgroup of children with DCD who have greater atypical sensory modulation compared to TD children. While these differences are not as severe or prevalent as those seen in ASD (except for bystander), they are notable nevertheless for their implications on subtyping and individualized interventions. Thus, when assessing individuals with DCD, it is important to screen for challenges in sensory processing and include interventions to mitigate these challenges.

#### 4.1.2. Anxiety and Depression

The DCD and TD groups did not differ on parent-reported measures of anxiety and depression/emotional withdrawal, and both groups were significantly lower than the ASD group. Consistent with other studies [62,63,126,127], parents of participants in all three groups reported levels of anxiety in the clinical range, with ASD reporting the highest incidence, followed by DCD and then TD. 

#### 4.1.3. Empathy

There were no group differences in empathic processing except in children with ASD showing higher personal distress than TD peers. This is contrary to prior evidence showing that children with ASD and those with poor motor abilities have less cognitive and/or emotional empathy [19,81,128]. In our studies, we used a self-reported measure of empathy (IRI) compared to parent-reported measures utilized in other studies [81], which may indicate that children are more empathetic than their parents report.

Personal distress is often considered a maladaptive empathy response associated with depression, rumination, and negative self-image [73,129], and increased personal distress in ASD is consistent with a prior study from our group with lower sample sizes [73]. As we have already discussed the nuances of empathy in ASD in our previous paper (interactions with alexithymia and interoceptive processing being key modulators to empathy in ASD) [73], we do not expand on this finding here, only noting that our results here do not show support for cognitive or emotional empathy differences in the DCD group. 

#### 4.1.4. Alexithymia

The DCD and TD groups did not differ in alexithymia; however, the ASD group scored significantly higher on alexithymia than both groups, consistent with our prior research and that of others [68,73]. To our knowledge, this is the first study to report that children with DCD have scores on alexithymia comparable to those of their TD peers. 

#### 4.1.5. Other Social Emotional Measures (CBCL Syndromes & NEPSY)

The DCD group showed significantly more difficulties with social problems, attention problems, and thought problems on the CBCL than the TD group but significantly less than the ASD group. However, the DCD group was not significantly different from the TD group on other social emotional measures such as social competencies, ToM, and somatic complaints. There were no significant differences between any groups on Affect Recognition, which conflicts with results from prior studies in which ASD and DCD groups performed worse than TD peers on face processing [130]. However, Affect Recognition in the ASD group may depend on individual differences in other skills, such as imitation ability [131]. Further, discrepant findings may be due to the utilization of different measures of Affect Recognition in different studies (e.g., NEPSY used here compared to the battery of face processing tasks used in Bruce et al. [132]) or the differences in the participants’ ages (7–10 years in Sumner et al. [130] compared to 8–17 used here). Nevertheless, these results indicate that. compared to TD groups, children with DCD have social emotional difficulties (social, attention, and thought problems) that should be considered in diagnosis and interventions. 

#### 4.1.6. Behavior

The DCD group fell between the ASD and TD groups on school competencies and total competencies as reported on the CBCL. Rule-breaking behavior was not significantly different among ASD and DCD children, but in both clinical groups parents reported more rule-breaking behaviors than for TD children, consistent with findings from studies on behavior in children with DCD and ASD [133,134,135]. While this data is compelling, we note that it may have resulted from the exclusion criteria of ADHD in the TD group that were not applied to the DCD or ASD groups (both of which showed more ADHD symptoms than the TD group), which is likely to contribute to decreased school performance and increased conduct problems [2,49,136]. Therefore, it is not surprising that the DCD group performed worse on CBCL competencies compared to the TD group in the current study. Future studies comparing children with DCD to children with ADHD without comorbid DCD could shed more light on this topic. 

Interestingly, TD and DCD children were comparable in social competencies and competencies in activities (including sports performance), contrary to findings from prior studies that report poorer social interactions, low engagement in organized sports, and more internalizing and externalizing problems in children with DCD than in TD peers [53,137,138,139]. 

All subscales of repetitive behaviors on the RBS-R were significantly (or nearly significantly for self-injurious behaviors) more common in the ASD group compared to the DCD and TD groups. The TD and DCD groups did not significantly differ in the amounts of total repetitive behaviors by parent report. Wigham et al. (2014) [140] hypothesized that repetitive behaviors are employed as a strategy to manage anxiety, which can be explored in future studies. 

### 4.2. Correlations

#### 4.2.1. Social Skills

Decreased social skills, as measured by the SRS-2, were positively correlated with sensory over-responsivity in the ASD group and trended towards significance in the DCD group. While such relationships have previously been shown in the ASD group [141], to our knowledge these are the first data indicating that similar patterns may emerge in the DCD group with larger data sets. This result is consistent with our hypothesis that the DCD group would show correlations between sensory processing and social functioning similar to the ASD group.

#### 4.2.2. Anxiety and Depression

Consistent with previous research [142,143,144], anxiety correlated with sensory over-responsivity in all groups. Post hoc analyses indicated that this was particularly driven by the social anxiety subscore in the DCD group and separation anxiety in the TD group (see Table 3), while all subscores contributed similarly to anxiety in the ASD group. Depression and emotional withdrawal were also significantly positively correlated with sensory over-responsivity in all groups. Combined with the similar finding with anxiety, this emphasizes the need to better understand the relationship between social emotional and sensory factors in these populations. It has been suggested that subtyping of groups based on sensory profiles can aid in more targeted and effective individualized treatments for mental health [67]. In an earlier paper, Green and Ben-Sasson [59] examined the relationship between anxiety and sensory over-responsivity in ASD and proposed three possible theories to explain the association between anxiety and sensory over-responsivity: (a) anxiety causes sensory over-responsivity; (b) sensory over-responsivity causes anxiety; or (c) sensory over-responsivity and anxiety are causally unrelated but are associated through diagnostic overlap. These same issues need to be examined in children with DCD. 

#### 4.2.3. Empathy

Here, we found that increased sensory over-responsivity was related to poorer perspective taking (cognitive empathy) skills in the ASD group and poorer empathic concern (emotional empathy) for the DCD group. Difficulty in cognitive empathy, in particular perspective taking, is a common feature of ASD [145], and relationships with sensory sensitivities have been previously reported [146]. 

To our knowledge, this is the first report on DCD showing that decreased empathic concern is related to increased sensory sensitivities. To delve into this unexpected result further, we conducted a post hoc analysis and found that this relationship was driven by tactile over-responsivity (r = −0.597, *p* < 0.01). Indeed, a prior study showed a relationship between motor impairment in DCD and tactile sensitivity [43]. This may indicate that more tactile sensitivity, common in individuals with motor impairments, leads to overwhelming sensory responses, which may block simulating other people’s emotional experiences, thus decreasing empathic concern. This remains highly speculative, and thus should be explored further in future research. Nevertheless, this may indicate the importance of screening and interventions for sensory modulation that may impact social emotional behaviors in DCD. 

#### 4.2.4. Alexithymia

Interestingly, we found that in the ASD group, which shows significantly more alexithymia than both TD and DCD groups, increased sensory over-responsivity was related to increased alexithymia. Prior studies have shown relationships between sensory processing and alexithymia in ASD; however, findings have been mixed, with some reporting correlations between under-responsivity and alexithymia [65,69,70,147]. The varied results from these studies and our own findings show the nuances of sensory processing in ASD groups and how they can differentially relate to the presence of alexithymia. There were no correlations between alexithymia and sensory processing for the TD or DCD groups.

#### 4.2.5. Other Social Emotional Measures (CBCL Syndromes & NEPSY)

Theory of Mind and Affect Recognition were not correlated with sensory processing in any group. The DCD and TD groups did not display significant correlations between sensory over-responsivity and the other social emotional measures on the CBCL. However, social problems, thought problems, and attention problems were positively correlated with sensory over-responsivity in the ASD group. These correlations are supported by previous studies [51,148].

#### 4.2.6. Child Behavior Checklist

Post hoc analyses on CBCL subscores show significant correlations between sensory over-responsivity and aggressive behavior in the ASD group and trending significance in the DCD group. These relationships are consistent with a study by Mazurek et al. [149] in which sensory problems were strongly associated with aggression in children and adolescents with ASD and may reflect the sympathetic response to sensory overload [150]. Although this has not been studied in DCD, children with SPD displayed more aggression towards others than typically developing peers in a study by Cosbey and colleagues [45]. This suggests that sensory processing may be a contributor to aggressive behavior, and future research should focus on further deciphering this relationship. 

#### 4.2.7. Repetitive Behaviors

Consistent with prior studies, the ASD group showed correlations between sensory over-responsivity and repetitive behaviors [47,51,82,83,151,152]. The DCD group showed a similar pattern with the self-injurious, sameness, and total repetitive behaviors. Atypical sensory processing has been established as a risk factor for self-injurious repetitive behaviors in children with ASD [153]; however, this is the first instance reported in a DCD sample. 

It has been hypothesized that repetitive behaviors are employed as a strategy for self-regulation to reduce anxiety in children with ASD [140,151]. Although we did not explore whether anxiety modulates the strength between repetitive behaviors in these groups, future research should focus on this in DCD and TD populations. 

#### 4.2.8. Sensory Over-Responsivity and Motor Skills

In the DCD group only, as sensory over-responsiveness decreased, motor skills increased. This result expands upon prior studies on participants with SPD showing a relationship between motor problems and sensory sensitivity [8,43], suggesting that relationships between sensory modulation and motor abilities can be seen in both DCD and SPD. Therefore, in developmental disorders where the primary deficit is in motor (DCD) or sensory processing (SPD), sensory and motor processing are related. In ASD, there is wider phenotypic heterogeneity, and relationships between sensory and motor behavior are less observed; perhaps both are related to social processing instead [84,85].

### 4.3. Limitations

The greatest limitation of this study is the small sample size of children with DCD/dyspraxia. We had a few findings that were trending towards significance for the DCD group (correlations between sensory over-responsivity and social skills, aggressive behavior, and emotional withdrawal/depression, and a group difference between DCD and ASD for self-injurious behaviors) that may have been significant with a larger sample size, as indicated by the large effect sizes. Further, some of our inclusion criteria limit the generalizability of these results, such as right-handedness, FSIQ > 75, and no psychological diagnosis or neurological disorders (TD group only). Thus, larger studies with more heterogeneous groups are needed to test generalizability of the current results. In addition, for correlational analysis, we used the SensOR Inventory, a measure of sensory over-responsivity; therefore, we cannot make any claims relevant to sensory under-responsivity. We suggest that future research should focus on under-responsivity to better understand the entire spectrum of sensory sensitivities in DCD. Finally, all measures were self- or parent-reported, which leaves an opportunity for future research to explore these relationships with performance-based or observational measures.

## 5. Conclusions

Our results indicate that abnormalities in sensory processing, including both discrimination and modulation, should be considered in understanding the possible phenotypes of DCD and developing individualized interventions. Further, given the correlation between sensory modulation scores and social emotional skills, interventions focusing on sensory modulation in DCD (and ASD) may also have the added benefit of improving social skills, mental health, and behavior [154]. For children with DCD, sensory processing is associated differently with social emotional measures, behavior, and motor skills than for children with ASD and TD children, and further studies are needed to better understand these differences. To our knowledge, this is the first study of this scope on the DCD population. Future research on a larger sample of this population will allow us to characterize it more fully.

## Figures and Tables

**Figure 1 brainsci-12-01171-f001:**
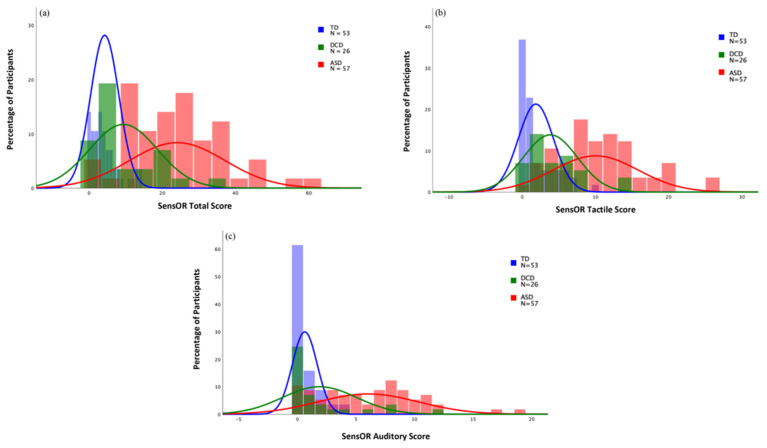
Histograms for SensOR total score and subscores. (**a**) SensOR total score; (**b**) SensOR tactile score; (**c**) SensOR auditory score.

**Figure 2 brainsci-12-01171-f002:**
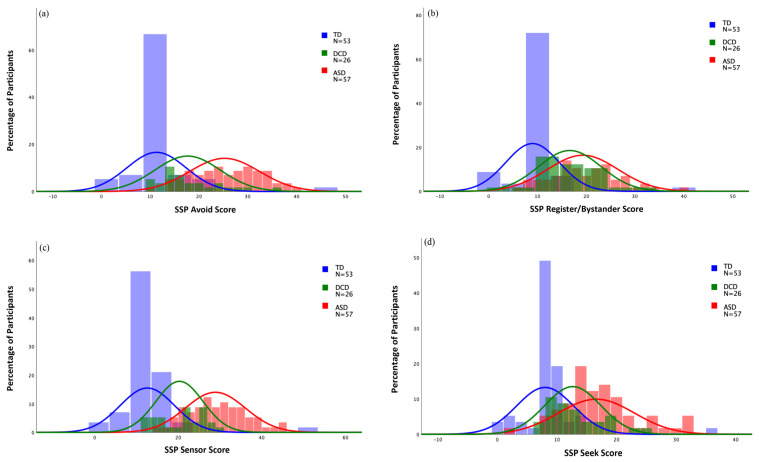
Histograms for SSP-2 sensory response patterns. (**a**) Avoider; (**b**) bystander; (**c**) sensor; (**d**) seeker.

**Figure 3 brainsci-12-01171-f003:**
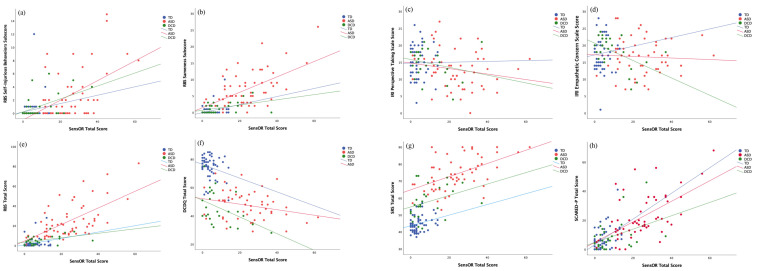
Scatterplots depicting significant correlations. (**a**) SensOR total and RBS-R self-injurious behavior; (**b**) SensOR total and RBS-R sameness; (**c**) SensOR total and IRI perspective taking; (**d**) SensOR total and IRI empathetic concern; (**e**) SensOR total and RBS-R total score; (**f**) SensOR total and DCDQ total score; (**g**) SensOR total and SRS total score; (**h**) SensOR total and SCARED-P total score.

**Table 1 brainsci-12-01171-t001:** The means, standard deviations, *p* values, and partial eta squares for each group with between-group comparisons for demographic and variable information controlling for age, gender, and FSIQ.

	DCD *N* = 26		ASD *N* = 57		TD *N* = 53		DCD:TD	DCD:ASD	TD:ASD	
	Mean	SD	Mean	SD	Mean	SD	*p*	*p*	*p*	Partial Eta Squared
Sex (sum M)	15	-	44	-	31	-	0.041 ^d^	0.041 ^d^	0.041 ^d^	0.198 ^d^
Age	11.75	2.31	11.89	2.29	11.75	2.13	1.00	1.00	0.974	0.001
Full-Scale IQ	109.69	17.13	107.51	16.82	118.28	13.74	0.083	1.00	<0.001	0.094
DCD/ASD > TD or TD > DCD/ASD										
SSP-2 Bystander	16.62	6.41	19.28	7.24	8.98	5.47	<0.001 **	0.225	<0.001 **	0.355
DCDQ Total ^c^	45.36	11.81	47.47	9.98	74.32	7.95	<0.001 ***	1.00	<0.001 **	0.658
DCD between ASD and TD: ASD > DCD > TD or TD > DCD > ASD										
SSP-2 Seeker	12.65	4.70	16.61	6.38	8.00	4.78	<0.001 **	0.015 *	<0.001 **	0.318
SSP-2 Avoider	17.65	6.70	25.23	7.15	11.28	6.06	<0.001 **	<0.001 **	<0.001 **	0.463
SSP-2 Sensor	20.31	5.64	28.89	7.18	12.64	6.49	<0.001 **	<0.001 **	<0.001 **	0.513
CBCL Social Problems	58.92	7.12	64.44	9.80	51.55	2.85	<0.001 ***	0.002 **	<0.001 ***	0.393
CBCL Attention Problems	58.23	5.72	65.25	11.78	51.34	2.24	0.002 **	<0.001 ***	<0.001 ***	0.379
CBCL Thought Problems	56.50	6.69	65.11	9.61	51.98	3.29	0.024 *	<0.001 ***	<0.001 ***	0.400
CBCL Total Competencies	43.23	10.90	34.70	7.51	50.43	10.06	0.015 *	<0.001 ***	<0.001 ***	0.323
CBCL School Competencies	45.27	8.27	40.32	8.00	52.94	3.47	<0.001 ***	0.004 **	<0.001 ***	0.380
ASD > DCD/TD or DCD/TD > ASD										
SensOR Total	9.38	9.31	24.07	13.04	4.26	3.87	0.128	<0.001 ***	<0.001 ***	0.441
SensOR Tactile	3.85	3.70	10.02	5.84	1.87	2.40	0.105	<0.001 **	<0.001 **	0.389
SensOR Auditory	1.88	3.18	6.04	4.31	0.62	1.06	0.141	<0.001 **	<0.001 **	0.360
RBS Stereotyped	1.12	1.45	3.51	2.89	0.08	0.33	0.121	<0.001 ***	<0.001 ***	0.354
RBS Compulsive	0.54	0.81	3.67	3.98	0.62	1.43	1.00	<0.001 **	<0.001 ***	0.226
RBS Ritualistic	0.69	1.10	4.89	3.53	0.42	1.31	1.00	<0.001 ***	<0.001 ***	0.422
RBS Sameness	1.42	1.68	7.07	5.35	0.45	1.55	0.699	<0.001 ***	<0.001 ***	0.404
RBS Restricted	0.58	1.14	2.88	2.36	0.09	0.35	0.843	<0.001 ***	<0.001 ***	0.352
RBS Total	5.46	4.51	24.67	17.45	2.11	5.25	0.710	<0.001 ***	<0.001 ***	0.423
CBCL Anxious/Depressed	53.69	6.28	61.91	10.65	52.42	4.42	1.00	<0.001 ***	<0.001 ***	0.254
CBCL Withdrawn/Depressed	54.96	6.70	62.70	10.19	51.70	2.64	0.153	<0.001 ***	<0.001 ***	0.311
CBCL Somatic Complaints	55.00	6.40	61.56	10.14	52.83	4.41	0.843	<0.001 ***	<0.001 ***	0.215
CBCL Social Competencies	45.92	10.03	36.38	8.87	49.09	8.93	0.772	<0.001 ***	<0.001 ***	0.246
CBCL Aggressive Behavior	53.00	4.29	57.46	9.16	50.68	1.76	0.307	0.007 **	<0.001 ***	0.201
SCARED-P GAD	3.58	4.38	6.67	4.58	2.25	2.73	0.395	0.002 **	<0.001 ***	0.219
SCARED-P Social Anxiety	2.38	2.80	5.74	4.03	2.64	3.12	1.00	<0.001 ***	<0.001 ***	0.190
SCARED-P Separation Anxiety	1.35	2.12	4.21	3.83	1.11	1.80	1.00	<0.001 ***	<0.001 ***	0.219
SCARED-P School Avoidance	0.65	1.16	1.40	1.90	0.36	0.762	1.00	0.046 *	<0.001 ***	0.107
SCARED-P Panic	1.04	2.46	3.75	4.33	0.62	1.735	1.00	<0.001 ***	<0.001 ***	0.192
SCARED-P Total ^b^	9.00	10.68	21.77	15.14	6.87	7.64	1.00	<0.001 ***	<0.001 ***	0.278
NEPSY-ToM	24.92	2.51	22.42	3.43	25.47	1.98	1.00	<0.001 **	<0.001 **	0.154
Alexi-Describing Feelings	0.62	0.39	1.01	0.43	0.73	0.48	1.00	<0.001 **	0.002 *	0.132
Alexi-2-factor total	6.88	4.19	9.72	4.65	7.11	4.68	1.00	0.022 *	0.010 *	0.083
ASD > TD only										
RBS Self-Injurious	1.12	1.86	2.56	3.61	0.45	1.74	0.907	0.051 ^t^	<0.001 ***	0.119
CBCL Rule-Breaking Behavior	53.50	5.13	55.23	6.71	51.00	1.93	0.084	0.257	<0.001 ***	0.146
IRI Personal Distress ^a^	13.42	5.0	14.59	4.69	12.40	5.15	1.00	0.432	0.021 *	0.056
TD > ASD only										
CBCL Activities Competencies	45.42	11.56	40.65	8.94	48.94	10.10	0.406	0.125	<0.001 ***	0.122
No significant differences between groups										
IRI Perspective Taking ^a^	14.73	4.40	12.79	5.58	14.81	4.70	1.00	0.156	0.088	0.046
IRI Fantasy Scale ^a^	17.38	5.32	16.61	5.67	16.77	5.41	1.00	1.00	1.00	0.004
IRI Empathetic Concern ^a^	18.23	4.55	16.63	5.26	17.72	4.73	1.00	0.809	1.00	0.009
NEPSY-Affect Recognition	27.88	3.04	26.26	3.43	28.25	2.94	1.00	0.378	0.140	0.036
Alexi-Identifying Feelings	0.54	0.41	0.67	0.46	0.49	0.39	1.00	0.571	0.138	0.032

^a^ = missing 1 ASD participant; ^b^ = missing 1 TD participant; ^c^ = missing 1 DCD participant; ^d^ = Cramer’s V; * = *p* < 0.05, ** = *p* < 0.01; *** = *p* < 0.001; t = *p* < 0.065; M = Male; IQ = Intelligence Quotient; SSP = Short Sensory Profile; DCDQ = Developmental Coordination Disorder Questionnaire; CBCL = Child Behavior Checklist; SensOR = Sensory Over-Responsivity Inventory; RBS = Repetitive Behaviors Scale; SCARED-P = SCARED Parent; GAD = Generalized Anxiety Disorder; NEPSY = Neuropsychological Assessment; ToM = Theory of Mind; Alexi = Alexithymia; IRI = Interpersonal Reactivity Index.

**Table 2 brainsci-12-01171-t002:** Within-group partial correlation (r values) between sensory over-responsivity (SensOR total score) and social emotional measures, repetitive behaviors, and motor skills while controlling for age, sex, and IQ by group.

	SCARED-P	SRS	IRI				NEPSY		RBS							DCDQ
Group	Total	Total	PT	EC	FS	PD	AR	ToM	Stereo	SI	Comp	Rit	Same	Rest	Total	Total
DCD	0.435 *	0.395 ^t^	0.062	−0.601 **	−0.040	−0.061	0.322	0.194	0.098	0.472 *	0.117	0.321	0.575 **	0.189	0.570 **	−0.433 *
ASD	0.667 ***	0.575 ***	−0.316 *	−0.022	−0.008	−0.050	0.088	0.103	0.371 **	0.560 ***	0.485 ***	0.561 ***	0.586 ***	0.386 **	0.631 ***	−0.249
TD	0.364 *	0.184	0.042	0.157	−0.103	0.085	0.012	−0.085	−0.018	0.068	0.103	0.220	0.233	−0.041	0.167	−0.270

* = *p* < 0.05; ** = *p* < 0.01; *** = *p* < 0.001; t = *p* < 0.065; SensOR = Sensory Over-Responsivity Inventory; SCARED-P = SCARED Parent; SRS = Social Responsivity Scale; IRI = Interpersonal Reactivity Index; PT = perspective taking; EC = empathetic concern; FS = fantasy scale; PD = personal distress; NEPSY = Neuropsychological Assessment; AR = Affect Recognition; ToM = Theory of Mind; RBS = Repetitive Behaviors Scale; Stereo = stereotyped; SI = self-injurious; Comp = compulsive; Rit = ritualistic; Same = sameness; Rest = restricted; DCDQ = Developmental Coordination Disorder Questionnaire.

**Table 3 brainsci-12-01171-t003:** Within-group partial correlation (r values) between sensory over-responsivity (SensOR total score) and anxiety subscores, alexithymia, empathy, and the CBCL, while controlling for age, sex, and IQ by group.

	SCARED-P Subscores					Alexithymia			CBCL											
Group	GAD	Panic	Soc Anx	Sep Anx	Sch Avoid	ID	Desc	2 Fact	ActCom	SchCom	SocCom	Total Com	Anx/D	W/D	Som Cx	Soc Px	ThPx	AttPx	RB Bx	AgBx
DCD	0.321	0.381	0.544 **	0.230	0.281	0.049	0.181	0.133	−0.090	0.310	0.082	0.014	0.448 *	0.584 **	0.039	0.238	0.245	0.121	0.256	0.403 ^t^
ASD	0.531 **	0.539 **	0.468 **	0.646 **	0.494 **	0.185	0.303 **	0.265 ^t^	0.157	−0.249	−064	0.021	0.506 ***	0.454 ***	0.503 ***	0.598 ***	0.613 ***	0.505 ***	0.364 **	0.299 *
TD	0.240	0.314	0.218	0.469 ***	−0.027	0.100	−0.169	0.026	−0.010	0.026	0.060	0.019	0.364 **	0.418 **	0.252	0.268	0.064	−0.014	0.089	−0.088

* = *p* < 0.05; ** = *p* < 0.01; *** = *p* < 0.001; t = *p* < 0.065; SensOR = Sensory Over-Responsivity Inventory; SCARED-P = SCARED Parent; GAD = Generalized Anxiety Disorder; Panic = panic disorder; Soc = social; Anx = anxiety; Sep = separation; Sch = school; Avoid = avoidance; ID: identifying feelings; Desc = describing feelings; Fact = factor; CBCL = Child Behavior Checklist; Act = activities; Com = competencies; D = depressed; W = withdrawn; Som = somatic; Cx = complaints; Th = thought; Px = problems; Att = attention; RB = rule breaking; Bx = behavior; Ag = aggressive.

## Data Availability

Not applicable.

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
