# Peer review of "Sensory Modulation in Children with Developmental Coordination Disorder Compared to Autism Spectrum Disorder and Typically Developing Children"

_brainsci, 2022, doi:10.3390/brainsci12091171_

Round 1

Reviewer 1 Report

Thank you for providing me with the opportunity to review your work. Congratulations on this outstanding work! 

I have a few comments:

1. It would be better to format the manuscript to make it appear more structured and easier to follow. For example, in the discussion section, the subheadings appear as if they are just independent words inserted randomly. Subheadings need proper formatting to make them easier to understand.

2. Conclusion section should not cite references.

Reviewer 2 Report

I have various methodological reservations regarding the submitted manuscript, especially regarding sampling and statistical methods.

I lack a justification of the sample size (optimal sample size). No alpha level is defined, nor is there a primary research question.

A large number of presumably highly correlated variables are examined for significant mean differences without any control for alpha error (e.g., Bonferoni).  The comparisons are multiple.

When comparing two groups, the effect size for pairwise comparisons is missing.

Some values in the table do not appear to be correct, for example, the p-values for IRI FS. Here the results should be checked again.

The layout of the tables should be adjusted. Some tables are not easy to read, see table 2

The correlation coefficients in Table 3 could be checked for significant differences.

It was not clear to me while reading the paper why the sizes of the three groups are so different. Especially the group DCD, which is in focus, is weak with 26 persons. The authors do discuss this in the limitations, but there would need to be a broader discussion of the causes here.

Were the prerequisites of the statistical procedures checked. Some of the measurement instruments are likely to yield values that have strong ceiling or floor effects. In this case, non-parametric methods would have to be used due to the violations of the preconditions.

Overall, there is a question about the primary research question. A variety of characteristics were collected in three groups. It would probably make more sense to use, for example, a stepwise discriminant analysis to determine the most statistically significant differences between the three groups rather than fishing for significance with multiple testing here.

It is not clear whether informed consent was obtained from the children and the parents.

Reviewer 3 Report

Thank you for the opportunity to review this interesting manuacript. The topic "sensory modulation in children with Developmental Coordination Disorders" is actual and  of interest. The topic has been well structured and presented. The decision to include people with Autism is appropriate. The introduction, even if long, presented the topic properly. The methodology is very well structured. Results are well supported by tables and figures. Discussion is complete. All the limits of the study are presented in an appropriate paragraph.

Considering all these aspects, I can only suggest this manuscript for publication.

Round 2

Reviewer 2 Report

The power calculation should be within the manuscript.

The correlation coefficients clacluated within the groups should compared for significant differences.
